# Effects of Short-Term Low Energy Availability on Metabolism and Performance-Related Parameters in Physically Active Adults

**DOI:** 10.3390/nu17020278

**Published:** 2025-01-14

**Authors:** Jana Nolte, Marius Kirmse, Markus de Marées, Petra Platen

**Affiliations:** Department of Sports Medicine and Sports Nutrition, Ruhr University Bochum, 44801 Bochum, Germany; marius.kirmse@rub.de (M.K.); markus.demarees@rub.de (M.d.M.); petra.platen@rub.de (P.P.)

**Keywords:** energy deficiency, caloric restriction, REDs, metabolic adaptations, athlete nutrition

## Abstract

Background/Objectives: Low energy availability (LEA) can cause impaired reproductive function, bone health issues, and suppressed immune function, and may result in decreased performance and overall health status. The purpose of this study was to investigate adaptions of body composition, blood status, resting metabolic rate, and endurance performance to gain more comprehensive insights into the symptoms of LEA and the adaptive effects in the athlete population (active women (n = 11) and men (n = 11)). Methods: Three treatments were defined as 45 (EA45, control), 30 (EA30), and 10 (EA10) kcal/kg FFM/day and randomly assigned. Pre- and post-intervention measurements were performed through blood sampling, bioelectrical impedance analysis, resting metabolic rate measurement, the oral glucose tolerance test (OGTT), and the incremental endurance test to exhaustion. Results: There was a significant reduction in body weight and fat mass in EA10 compared to EA45 (*p* ≤ 0.05). Blood serum levels were altered in triglyceride, uric acid, and creatinine concentrations in EA10 compared to EA45 (*p* ≤ 0.05). Furthermore, blood glucose was still accumulated after 120 min during OGTT in EA10 compared to EA45 (*p* ≤ 0.05). The respiratory exchange ratio was reduced during submaximal stages of the incremental treadmill test to exhaustion without influencing performance output after treatment EA10 (*p* ≤ 0.05). However, the resting metabolic rate did not change (*p* > 0.05). Conclusions: In conclusion, this short-term study indicates that energy restriction can lead to several metabolic-related adaptations, which suggests that the availability and regulation of glucose and fats are significantly influenced after only five days of LEA in physically active women and men. Future research should focus on longer exposures of LEA and sex-specific comparisons (including the menstrual cycle) on LEA symptoms.

## 1. Introduction

Low energy availability (LEA) is the underlying cause of the relative energy deficiency in sports (REDs) and its subset the female athlete triad (the triad). The triad is defined as a medical condition of LEA with or without disordered eating, menstrual dysfunction, and low bone mineral density [1]. REDs provides a more inclusive framework that considers the broader spectrum of potential health and performance consequences resulting from inadequate energy availability in both male and female athletes [2]. This syndrome is common in weight-sensitive sports, like aesthetic sports (such as gymnastics or figure skating) and sports with weight classes (such as combat sports or rowing) [3]. It is sex-inclusive and is defined as “impaired physiological function including, but not limited to, metabolic rate, menstrual function, bone health, immunity, protein synthesis, and cardiovascular health caused by relative energy deficiency” [4] (p. 1). This insidious condition, which results from an imbalance between energy expenditure (EE) and energy intake (EI), has been increasingly recognized as a pervasive issue affecting elite athletes across various disciplines. LEA not only undermines an athlete’s physical well-being but also exerts a profound impact on their performance and overall health.

Two concepts try to define the homeostatic state of energy. In the field of dietetics, the concept of energy balance is often used to cover total energy expenditure (TEE), which summarizes all work of the body’s physiological systems in one day (Energy balance = EI − TEE) [5]. Energy availability assumes that energy from food is consumed for several basic physiological processes, including cell maintenance, thermoregulation, growth, reproduction, immunity, and locomotion [6]. In exercise science, it is therefore defined as the amount of energy (calories (kcal) per kilogram (kg) of fat-free mass (FFM) per day) available for physiological processes, health, growth, and activities of daily living after subtracting the energy used for exercise and sporting activities (EEE: exercise energy expenditure) [4,7].$$Energy\;availability(kcal/{kg}_{FFM}/day)=\frac{EI-EEE}{FFM}$$

Since 2014, the International Olympic Committee (IOC) outlined the immediate and prolonged effects of REDs in many physiological systems, which adversely affects both health and performance [4]. Studies have found symptoms of LEA comparable to REDs, such as decreasing performance, exercise response, coordination, and concentration, and increasing irritability, and depression in women [8]. Similar findings have also been described for male athletes [9]. LEA symptoms are not limited to otherwise healthy young athletes; LEA can also occur in women and men of any level of physical activity and any age group [10].

Overall, it can perturb the endocrine system, leading to hormonal dysregulation, among other things. From 1994 to 2006, Anne Loucks and her team published a series of papers on the effects of LEA in healthy, sedentary women. They tested—among other parameters—luteinizing hormone (LH) pulsatile profiles and different metabolites in regularly menstruating women and examined energy availabilities of 45, 30, and 10 kcal/kg FFM/day to define a threshold at which LEA occurs [7,11,12,13]. Besides hormonal effects, 4 days of LEA below 30 kcal/kg FFM/day reduced 24 h mean plasma glucose concentrations, increased fasting morning ß-hydroxybutyrate, suppressed leptin, and linearly suppressed the insulin concentration by restricting energy availability in sedentary women [11].

LEA disrupts this finely tuned metabolic machinery, setting off a cascade of effects that compromise both short-term performance and long-term health [4]. When an athlete consistently consumes insufficient amounts of calories to meet his or her energy needs, the body enters a state of energy conservation. Prolonged LEA leads to metabolic adaptation, wherein the body strives to conserve energy to maintain essential functions [5]. The body seeks alternative energy sources. The glucose and fat metabolism appear to be a highly regulatory system when the energy supply is insufficient. “[W]orking muscle competes aggressively against the brain for plasma glucose” [11] (p. 309). If glucose is not available, the body accelerates the mobilization and catabolism of fat to maintain brain metabolism [14]. The body adjusts its physiological systems when there is not enough energy available. Within-day energy deficiency as well as prolonged caloric restriction lead to suppressed resting metabolic rate (RMR) [15,16,17], thereby conserving metabolic fuels for adequate energy supply to the brain and for other essential functions, such as cell maintenance, thermoregulation, and locomotion [18].

The reduction in RMR is an adaptive response, which aims to preserve energy stores and prioritize vital bodily functions. This is often accompanied by hormonal changes, such as decreased levels of thyroid hormones (T3 and T4) and leptin, while cortisol and ghrelin levels may increase [11,12]. These hormonal shifts can also contribute to reduced RMR as the body tries to conserve energy. LEA can also lead to muscle catabolism (breakdown) as the body seeks alternative energy sources. It impairs myofibrillar and sarcoplasmic muscle protein synthesis in trained females [19]. Since muscle tissue is metabolically active and contributes to RMR, a reduction in lean mass can further decrease RMR [19]. As a result of these adaptive responses, an athlete with LEA may experience a negative impact on their overall metabolic health and performance.

Previous reviews recommended further research on the metabolic effects of LEA in different populations and recommend the implementation of both short-term and long-term studies [10,20,21,22]. It is unclear whether such a threshold also exists in physically active, young women and men, as most of the studies were conducted with sedentary women [4]. Further methods to accurately measure energy availability should also be explored [4]. To address the research gap, this study aims to investigate the short-term effects of LEA in both sexes, which has not been extensively studied before. By including both physically active young women and men, this research seeks to provide a more inclusive understanding of LEA’s impact across sexes. Additionally, this study will explore the immediate metabolic and performance-related adaptations to LEA, thereby expanding upon previous research that primarily focused on long-term effects and sedentary populations.

The purpose of this study was to examine the metabolic and performance-related effects of five days of two levels of LEA using some of the methodologies and insights of Anne Loucks’ studies and transferring them to physically active young women and men to gain more comprehensive insight into the symptoms of LEA and the adaptive effects in this population. We hypothesized that after only five days of caloric restriction to 30 and/or 10 kcal/kg FFM/day, adaptations of body systems will occur in a condition of LEA.

## 2. Materials and Methods

### 2.1. Participants

Thirty young, active, and healthy women and men were recruited (number obtained from a prior power analysis) for the present study, while twenty-two successfully completed it. All participants were sports students, non-smokers, and without any history of eating disorders. All subjects signed a written informed consent document before participation, including a description of the nature of the study, the risks and benefits, and their ability to resign at any time. The study was positively evaluated by the local ethics committee (Faculty of Sport Science, Ruhr-University Bochum (EKS S 02/2022)) and conducted in accordance with the Declaration of Helsinki.

A detailed screening was performed at the beginning of the nutrition intervention. The medical history of the participants was obtained through interviews and questionnaires including health and menstrual status.

### 2.2. Study Design

We investigated the incremental dose–response effects of energy availability on several metabolic and performance-related parameters. The experimental design provided a five-day dietary intervention with three different levels of energy availability. Two treatments were characterized by reduced energy availability, defined as 30 kcal/kg FFM/day (EA30) and 10 kcal/kg FFM/day (EA10). The other treatment represents the control group (45 kcal/kg FFM/day (EA45)). Participants were randomly assigned to one of three groups using Microsoft Excel, with the gender distribution being the only controlled variable. Prior to the intervention, a three-day baseline measurement period was conducted to obtain information on participants’ dietary and exercise behavior. Subsequently, each participant took part in two pretests on days four and five. Blood sampling, bioelectrical impedance analysis, determination of the resting metabolic rate (RMR), and the oral glucose tolerance test (OGTT) were performed on pretest day one. On pretest day two, an incremental treadmill test to exhaustion was performed. On days six to ten, the five days of defined energy availability took place followed by a repeat of the identical test battery on days 10 and 11 (Figure 1).

### 2.3. Experimental Protocol

Participants had to arrive at 06:30 am in a fasted state (12 h of no eating and drinking) on the first day of pre- and post-testing. The bioelectrical impedance analysis (InBody 770, InBody Europe B.V., Eschborn, Germany) was performed to determine body weight and composition including FFM, fat mass, and total body water (no gold-standard tool (see Section 4.4)). Venous blood samples were taken to determine the blood serum levels of different metabolic parameters (glucose, triglycerides, cholesterol (LDL, HDL), uric acid, urea, and creatinine) (COBAS INTEGRA 400 plus, Roche Diagnostics International AG, Rotkreuz, Switzerland). Furthermore, the RMR was measured by spirometry (METAMAX^®^ 3B, CORTEX Biophysik GmbH, Leipzig, Germany) over 25 min with a maximum deviation of 5% in the respiratory quotient over 5 min. Finally, the OGTT was performed by drinking 300 mL of water, including 75 g of glucose. Capillary blood samples were taken from the earlobe before and 30, 60, 90, and 120 min after drinking the glucose mix to determine the blood glucose concentration (BIOSEN S_line, EKF-diagnostic GmbH, Barleben, Germany).

On the second day of testing, all participants had the same standardized breakfast (3 kcal/kg FFM, providing 62% of energy from carbohydrates, 14% from fat, and 24% from protein, composed of bread, jam, and bananas) two hours before an incremental treadmill test to exhaustion. The treadmill protocol started at 2.0, 2.4, or 2.8 m/s, depending on gender and estimated endurance capacity (4-mmol/l lactate threshold), and was increased by 0.4 m/s every 5 min. After three stages, the gradient was increased by 5% and the intensity was ramped up by 0.2 m/s every 30 s until exhaustion. Capillary blood samples were taken for the determination of lactate and glucose concentrations (BIOSEN S_line, EKF-diagnostic GmbH, Barleben, Germany) at rest, after stages one to three and at the end of the test, as well as 1, 3, and 5 min afterwards. Subjective perceived exertion (BORG-scale) (RPE) was assessed after each stage. Heart rate (Polar H10, Polar Electro GmbH, Büttelborn, Germany), oxygen uptake, and carbon dioxide output (METAMAX^®^ 3B, CORTEX Biophysik GmbH, Leipzig, Germany) were recorded continuously during the treadmill test to determine the submaximal respiratory exchange ratio (RER) over the last 30 s of each stage (1–3) and maximal oxygen uptake (V’O_2 max_).

### 2.4. Diet

The diet during the five-day intervention period was standardized concerning caloric content (according to group allocation) and macronutrient composition (providing 56% of energy from carbohydrates, 24% from fat, and 20% from protein) to stipulate the energy intake for each participant. Additional intake of water was allowed ad libitum. A nutrition diary was used three days before and during the five-day intervention to control the daily energy intake using the MyFitnessPal application (MyFitnessPal, Under Armour Inc., Baltimore, MD, USA). The application determines caloric intake and composition of macronutrients. This consumer app previously showed good relative validity and accuracy, especially for total energy intake [23]. Energy intake (kcal) was normalized to FFM.

### 2.5. Energy Expenditure

TEE and EEE were determined by the Bouchard Physical Activity Record over the whole pre-intervention and intervention period. This activity record identifies 9 types and groups of activities of different levels of intensity corresponding to 1.0 to 7.8 metabolic equivalents of tasks (MET), which are recorded in 15 min intervals [24]. The values were summed up daily and multiplied by the respective MET value. The mean of the baseline period allowed the estimation of habitual energy expenditure per day (kcal/kg FFM) in activities of daily living (TEE-EEE), and separately, that of the sporting activity (EEE). The results of the intervention period were used to examine both intra- and interindividual differences in energy availability.

To induce and control EEE during the five-day intervention, participants had to run at least 30 min per day on a flat terrain with a constant, individually determined velocity allowing for an exact calculation of individual energy expenditure during running. Any other sporting activity was prohibited during the intervention. Energy expenditure from running was calculated using the formula from Margaria et al. (1962) for steady-state running: 1 kcal/kg body weight/km [25].

### 2.6. Calculation of Energy Availability

Individual daily energy availability was analyzed as described above (EA = EI − EEE). EI was calculated considering individual FFM and the assigned group (EA45, EA30, and EA10). Additionally, the same number of calories consumed through running (EEE) was added to the daily EI.

The pre-intervention measurements of the MyFitnessPal App were used to estimate habitual EI (kcal/kg FFM/day). The intervention measurements were used to assess the participants’ compliance with their group. Furthermore, EI, TEE, and EEE were used for the calculation of energy availability and energy balance. The Bouchard Activity Record allowed for the determination of intra- and interindividual differences in daily activities that could influence the required energy availability.

### 2.7. Statistics

Statistical analyses were performed with JASP (Version 0.16.3) (JASP Team, Amsterdam, The Netherlands). The distribution of normality and homogeneity of variance were tested using the Shapiro–Wilk test and Levene’s test. All variables were analyzed with one-way analysis of variance (ANOVA) (delta (post − pre) × condition (EA45 vs. EA30 vs. EA10)). Post-hoc comparisons using Bonferroni-adjusted *p*-values were used to identify locations of significant effects. Cohen’s d was calculated using the mean difference and its standard deviation. The level of significance was set to *p* ≤ 0.05. All data are presented as means ± SD.

## 3. Results

### 3.1. Experimental Treatment

Twenty-two participants (f = 11, m = 11, age 24 ± 1 years, body mass 72 ± 12 kg, height 178 ± 8 cm) successfully finished all required measurements and were included in the statistical analysis (drop-out reasons: due to illness and injury, non-compliance with group criteria, and personal motivation). Baseline and intervention measurements can be found in Table 1. Neither EI, TEE, EEE, nor EA differed between the groups during the baseline period (*p* > 0.05) (Appendix A). Further, TEE and EEE did not differ between groups during the intervention period (*p* > 0.05). Energy intake, however, differed between groups (*p* ≤ 0.05) and led to the intended energy availability of approximately 45, 30, and 10 kcal/kg FFM/day, respectively. Energy balance was only achieved in EA45 with no differences between EI and TEE (*p* > 0.05).

### 3.2. Effects on Body Composition, Blood Serum Levels, and Metabolism

Table 2 documents the parameters of body composition. Post-hoc tests show that EA10 significantly reduced body weight compared to control (95% CI [−3.1 −1.0]; d = 2.8; *p* < 0.001) but not EA30 (95% CI [−1.7 0.3]; d = 0.9; *p* = 0.279). Moreover, EA10 reduced body weight more than EA30 (95% CI [−2.3 −0.4]; d = 1.9; *p* = 0.005). Further, fat mass was reduced in EA10 compared to EA45 after treatment (95% CI [−0.9 −1.8]; d = 1.5; *p* = 0.04). Figure 2 indicates a slight dose–response effect: the lower the energy availability, the higher the reduction in body weight, fat mass, FFM, and total body water.

The effects of the five-day energy availability treatment on blood serum levels are shown in Table 3. Concentrations of glucose, total cholesterol, HDL cholesterol, LDL cholesterol, and urea did not differ between groups after the treatment (*p* > 0.05). However, the triglyceride concentration (95% CI [−0.7 −0.1]; d = 1.6; *p* = 0.028) and creatinine concentration (95% CI [−0.7 −0.0]; d = 1.5; *p* = 0.047) decreased in EA10 compared to control. The uric acid concentration increased in EA10 (95% CI [15.3 108.9]; d = 1.9; *p* = 0.01) and EA30 (95% CI [8.8 97.5]; d = 1.6; *p* = 0.02) compared to EA45.

No significant differences could be found for RMR in kcal/day and kcal/kg FFM/day between groups, respectively (*p* > 0.05) (Appendix A). Figure 3 shows consistently lower median values of RMR for lower energy availability and a skewed distribution, indicating higher heterogeneity for the treatment effect in the EA10 group.

Glucose concentration was significantly accumulated during the OGTT after 120 min in EA10 compared to EA45 (95% CI [1.0 56.0]; d = 1.5; *p* = 0.05) and compared to EA30 (95% CI [3.6 55.0; d = 1.6; *p* = 0.028). Treatment effects could not be shown in the resting state after 30, 60, and 90 min, respectively (*p* > 0.05). Figure 4 maps the curves of glucose concentrations during the OGTT before and after treatment of each group.

### 3.3. Effects on Performance-Related Parameters

The completion of the treadmill test (achieved level), submaximal and maximal heart rate, submaximal RPE, and the courses of lactate and glucose concentrations showed no significant differences between groups after treatment, respectively (*p* > 0.05). Further, V’O_2 max_ absolute (L/min) and relative (mL/min/kg) did not differ between groups (*p* > 0.05). Submaximal RER of post-testing was significantly reduced at stage one in EA10 compared to EA45 (95% CI [−1.0 −0.0]; d = 1.8; *p* = 0.037), and at stage two in EA10 compared to EA45 (95% CI [−0.1 −0.0]; d = 3.5; *p* < 0.001) and EA30 (95% CI [−0.1 −0.0]; d = 2.4; *p* = 0.001). Submaximal RER did not differ at stage 3 between groups (*p* > 0.05). Pre-to-post comparisons of RER in each stage are shown in Figure 5.

### 3.4. Bouchard Activity Record for Intraindividual Comparison of Required Energy Availability

The value of TEE−EEE varied between 1977 and 2745 kcal among participants over 8 days (Appendix A). Further, the calculated required energy availability to cover the energy needs varied between 38.4 and 56.2 kcal/kg FFM/day. The average of the required energy availability of all subjects was 45.7 kcal/kg FFM/day.

## 4. Discussion

The key finding of this study is that five days of LEA (10 kcal/kg FFM/day) is associated with (1) reductions in body weight and fat mass, (2) altered blood serum levels (lower triglycerides and creatinine, as well as higher uric acid concentrations), and (3) impaired glucose metabolism, as evidenced by the increased glucose concentrations during OGTT and lower submaximal RER values during the incremental treadmill test. In contrast to our hypotheses, LEA appears to not affect RMR.

### 4.1. Effects on Body Composition, Blood Serum Levels, and Metabolism

It is well known that diets primarily focused on fat loss are driven by a sustained caloric deficit [26]. Caloric deficits over five days at 10 kcal/kg FFM/day had shown reductions in weight and fat mass. Based on dose–response effects between groups, which showed a decrease in weight and fat mass, a possible long-term decrease in fat mass could be expected [26]. The FFM, representing the highly active metabolic tissue of the body, did not change between treatments. However, energy requirements could adapt under energy restriction due to the adaptive thermogenesis (AT) of FFM, without reducing it [27]. AT is the FFM-independent reduction in resting energy expenditure after caloric restriction due to metabolic and endocrine adaptations at the tissue level [27,28]. Chronic and severe starvation is associated with a substantial decrease in the metabolic rate of FFM as cellular metabolism is significantly downregulated [29]. Our study suggests that muscle and cellular tissue breakdown due to catabolic processes occur after only five days of energy restriction. Cellular degradation, through increased protein catabolism associated with tissue damage, can be presumed by the significantly altered uric acid concentrations [30]. An increase in uric acid was noted after caloric restriction at 10 and 30 kcal/kg FFM/day. Further, lower creatinine levels in EA10 could indicate a catabolic state of muscle tissue where muscle breakdown occurs, if kidney function is intact [31]. Furthermore, impaired myofibrillar and sarcoplasmic muscle protein synthesis could be shown in trained females after 10 days of LEA [19]. Nonetheless, a decline in muscle mass can only be expected after five weeks of energy restriction [32].

AT could not be detected in altered RMR since it remained unchanged after five days of LEA, although it is known that a persistent state of LEA lowers RMR in males and females [15,17,19,27,29,33]. In women with amenorrhea, a decreased RMR was detected as a result of reduced metabolic activity at the tissue level due to metabolic and endocrine adaptations that are indicative of energy conservation under conditions of LEA [28]. Even a within-day energy deficiency is associated with metabolic disturbances in male endurance athletes [16]. The lack of significant changes in RMR leads to conflicting evidence. Due to the small sample size and the limitations of RMR measurement using spirometry, which is known for high variability, these results should be interpreted with caution. However, with longer exposure, standardized measurement procedures, and the suggested dose–response relationship indicated in this study, changes in RMR consistent with other metabolic alterations can generally be expected.

Nevertheless, the results of our study may indicate the presence of AT, as it is associated with a decrease in insulin secretion during the early stages of weight loss [27]. The decreased ability to metabolize glucose was shown through OGTT after a caloric restriction at 10 kcal/kg FFM/day. The increased blood glucose concentrations during OGTT can be explained by a reduced ability of muscle tissue and the liver to absorb glucose. Impaired glucose uptake into the tissue could be attributed to decreased insulin secretion [27] or the onset of insulin resistance. Insulin secretion is typically down-regulated in states of LEA to allow more substrate availability in males and females [18,34,35,36]. However, it has been reported that insulin sensitivity is increased in amenorrhoeic athletes due to higher insulin growth factor binding protein-1 (IGFBP-1) values [37]. In contrast, studies also found that insulin like growth factor 1 (IGF-1) and IGFBP-1 levels resulted in a reduced ratio of IGF-1/IGFBP-1, which could reduce the bioactivity and hypoglycemic effect of IGF-1 [18]. These results indicate lower insulin secretion on the one hand, and, on the other hand, a possibly impaired insulin sensitivity. Insulin controls a complex series of interlocking metabolic pathways that work together to instruct cells how and when to store energy [38]. Therefore, it is not possible to make a definite statement about the mechanism of impaired glucose uptake.

Moreover, the magnitude of glucoregulatory responses (decreased insulin/IGF-I action) is directly related to the degree of suppression of gonadotropin-releasing hormone (GnRH) and LH pulse frequency [18]. Potential dose–response effects of LEA on LH pulsatility were found to closely resemble those on plasma glucose and GnRH in young women [39]. However, we found no significant differences in plasma glucose levels that might indicate such a relationship.

In women, LEA is known to lead to disruption of the menstrual cycle, suppression of the reproductive systems, and functional hypothalamic amenorrhea as a result of the mechanism of energy conservation for processes more important than reproduction [17]. Energy conservation could not be proven in our short-term study, but after a short period of energy restriction, the use of alternative energy supplies becomes apparent.

### 4.2. Effects on Performance-Related Parameters

A shift in metabolism from decreased utilization of glucose to increased utilization of fats was evidenced by the results of RER under load. The treatment with a caloric restriction of 10 kcal/kg FFM/day showed significant shifts in RER mean values in stages one and two of the incremental treadmill test. Day-to-day differences were eliminated by serving the same breakfast to each group, so these effects can be considered as intervention effects. The lower RER values after treatment indicate depleted glycogen pools due to the rapid reduction in glycogen content within skeletal muscle and liver cell granules because of the low-caloric diet over several days and an additional running intervention [40,41]. Fat, as a metabolic substrate, is primarily used as an energy supplier, which is visible in the lower RER values. Low levels of triglycerides in blood serum in our study are indicative of the same effect [42]. We also hypothesize that glucose, as an important energy source, is increasingly used to supply the brain, and therefore glucose metabolism in the muscle cell is impaired due to glucose molecules not being freely available in the blood. The brain occupies a special hierarchical position in the organism and is separated from the general circulation by the blood-brain barrier [43]. Hence, the working muscles compete against the brain for plasma glucose [11]. Due to this mechanism, the brain is also called “the selfish brain” [43]. Brain energy expenditure is constant over a wide range of brain activity [44], whereas muscle energy expenditure depends on physical activity and can be down-regulated. A responsible mechanism, via the limbic–hypothalamus–pituitary–adrenal system, affects Adenosine triphosphate (ATP) concentration in the brain by locally stimulating glucose uptake across the blood–brain barrier and systematically inhibiting glucose uptake into muscle and adipose tissue [43]. Extreme stress situations (starvation, sports, and hormonal fluctuations) can also influence energy distribution [43]. The blood–brain barrier permeability of glucose can increase from the blood to the brain by more than 50% to satisfy brain energy needs [14]. Thus, a shift in RER values may be an indicator of the previously described mechanisms but should be urgently tested with more sensitive methods. LEA can result in complex metabolic changes, and the first measurable adjustments can occur after only five days of energy restriction.

We included physically active sports students as participants who are used to being physically active. Therefore, we allowed them to stay physically active; however, they were only allowed to run. Running speed and endurance capacity may influence the depletion of glucose stores. Nevertheless, the influence of physical activity on symptoms of LEA is well studied. Several findings regarding LEA in sedentary women are similar to those observed in highly trained athletes [7,18]. Loucks et al. (1998) showed no disruptive effects on LH pulsatility of intense exercise apart from the impact of its energy cost on energy availability [7]. Thus, nutritional deficiencies may be a common factor contributing to the development of various neuroendocrine-metabolic adaptations underlying both nutrition- and exercise-induced functional hypothalamic amenorrhea [18]. Reduced testosterone levels in men resulting from high volumes of aerobic exercise, known as the exercise-hypogonadal male condition, could also be associated with reduced energy intake [45]. Based on this, we assume that no special consideration needs to be given to the additional sporting activity.

Nonetheless, the influence of LEA on physical performance is questionable. Additional training may affect the kinetics of FFM in the early phases of weight loss treatments [40]. Hence, caloric restriction without exercise results in a reduction in FFM, whereas additional exercise does not affect FFM despite no change in net energy restriction [40,46]. This is where AT is most likely to occur due to lower requirements for ATP or lower energy requirements for ATP generation [32]. Physical activity under acute energy deficit leads to the increased activation of AMP-Activated Protein Kinase, which activates the uptake of substrates from plasma, lipolysis, mitochondrial function, and fat oxidation. These effects lead to increased cardiorespiratory capacity, increased mitochondrial oxidative capacity, decreased lipids in plasma, tissues, and cells, and improved insulin sensitivity [32]. Therefore, short-term energy restriction may have a positive effect on endurance performance [32], or at least does not have a negative effect [41], which is why running performance was not compromised in our study.

### 4.3. Energy Availability

Energy balance and energy availability try to define the relationship between energy intake and energy expenditure, but the risk of an undetectable negative energy balance over a longer period is very high. Various body systems and physiological processes regulate energy expenditure by slowing down to conserve energy. In a pathological state of a suppressed physiological system, an energy balance of zero can occur even though the body is suffering from permanent low energy availability [5,47]. Although this risk does not exist when considering the concept of energy availability, intraindividual differences must be mentioned. Not everyone needs exactly 45 kcal/kg FFM/day. In our study, we were able to determine TEE with and without exercise via a questionnaire. Differences could be found for the individual required energy availability. Both higher and lower required energy availability than 45 kcal/kg FFM/day were achieved. Body composition, as well as daily activities, have a significant influence. In the concept of energy availability, the energy expenditure of daily activities is considered a constant variable. This indicates that the general use of threshold concepts must be considered with caution because of interindividual differences. However, on average, our sample of twenty-two participants confirmed that they require 45 kcal/kg FFM/day to meet all required physiological processes and activities.

### 4.4. Strength, Limitations, and Future Directions

Our study provides valuable insights into the short-term metabolic adaptations to LEA in both men and women, highlighting significant changes in body composition, glucose regulation, and blood biomarkers. The randomized design with clearly defined energy availability groups strengthens the reliability of the findings. Our study benefits from its highly controlled nature: we could be confident in participants’ adherence to the diet due to joint development and implementation of the study, as well as the double check with calorie compliance. However, energy availability refers to the behavior of the participants, not to the cellular availability inside them. It is not under our control to affect the cellular availability of metabolic fuels, but we expect it [11].

These findings emphasize the importance of monitoring energy intake and expenditure to prevent health issues and performance decline in athletes.

The limitations of the study are (1) performing measurements that require a high degree of standardization: bioelectrical impedance analysis demonstrates comparable accuracy, but does not meet the gold standard [48] (future studies should consider using gold-standard tools (e.g., DEXA for body composition, doubly labeled water for energy expenditure); RMR measurements via spirometry had high variability of the measurement itself and the results, and the exclusion of some subjects due to not meeting criteria in the analysis may have had an impact on the outcome; (2) missing familiarization session, which is recommended for the RMR measurements and incremental treadmill test due to learning effects; (3) estimation of total energy expenditure: the Bouchard Activity Record was used to estimate the energy expenditure of the subjects and was not a direct measurement of it, therefore the results must be considered carefully; (4) determination of caloric intake: the underestimation of habitual energy intake by using food diaries with direct feedback is likely, and we did not consider “diet-induced thermogenesis”, which refers to 8–15% of total daily energy expenditure [26]; and (5) finally, the small sample size (n = 22) may limit the generalizability of findings. This limits the power of the study, and we were unable to account for sex differences and the menstrual cycle of women due to the small sample size.

This raises some prospects for follow-up studies. In exercising and non-exercising women, the role of the suppression of metabolic hormones due to LEA in the development of reproductive disorders is well established [7,12,17,35]. It is not surprising that the male reproductive system appears more robust to short-term LEA [36]. Future research should focus on long-term studies to assess the persistence of metabolic changes with prolonged LEA, and the impact of different training intensities, using appropriate measurements for EEE, and sex-specific comparisons (including the menstrual cycle), on LEA symptoms. Including standardized menstrual cycle monitoring, considering the hormonal differences between the cycle phases and between women and men, and ensuring that the test protocol and the intervention are standardized accordingly could give a more comprehensive understanding of the underlying mechanisms of LEA. Additionally, exploring the relationship between LEA and psychological factors like stress, eating behaviors, and mental health would offer a more comprehensive understanding of its effects on athletes.

The practical applications of this study are significant for coaches, nutritionists, and healthcare professionals, as they can use the insights to better identify early signs of LEA in athletes, such as changes in body composition, glucose regulation, and blood biomarkers. By monitoring short-term energy deficits, practitioners can help prevent long-term health complications, including impaired reproductive function, bone health issues, and performance decline. This study further emphasizes the importance of balancing energy intake and expenditure in training protocols to avoid metabolic disturbances and ensure the overall health and performance of athletes.

## 5. Conclusions

In summary, five days of reducing energy intake to an energy availability of 10 kcal/kg FFM/day resulted in decreased body weight and fat mass, decreased ability to uptake glucose into cells, and lower RER values during exercise. Collectively, this study suggests that metabolism is self-regulating in terms of glucose and fat availability to provide sufficient energy for essential body systems. A down-regulation of metabolism visible in the measurement of RMR can only be suspected and not proven in this study.

We aim to increase awareness among sports scientists, coaches, and athletes themselves about the critical importance of maintaining adequate energy availability for metabolic health.

## Figures and Tables

**Figure 1 nutrients-17-00278-f001:**
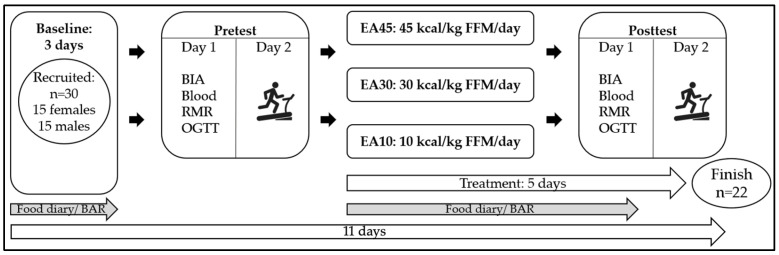
Systematic illustration of the study design. The study consisted of a three-day baseline period and a five-day intervention period, which were accompanied by a food diary and the Bouchard Activity Record (BAR) (grey arrows). BIA = bioelectrical impedance analysis; Blood = venous blood sampling; RMR = resting metabolic rate measurement; OGTT = oral glucose tolerance test; EA = energy availability; Pictogram = incremental treadmill test to exhaustion.

**Figure 2 nutrients-17-00278-f002:**
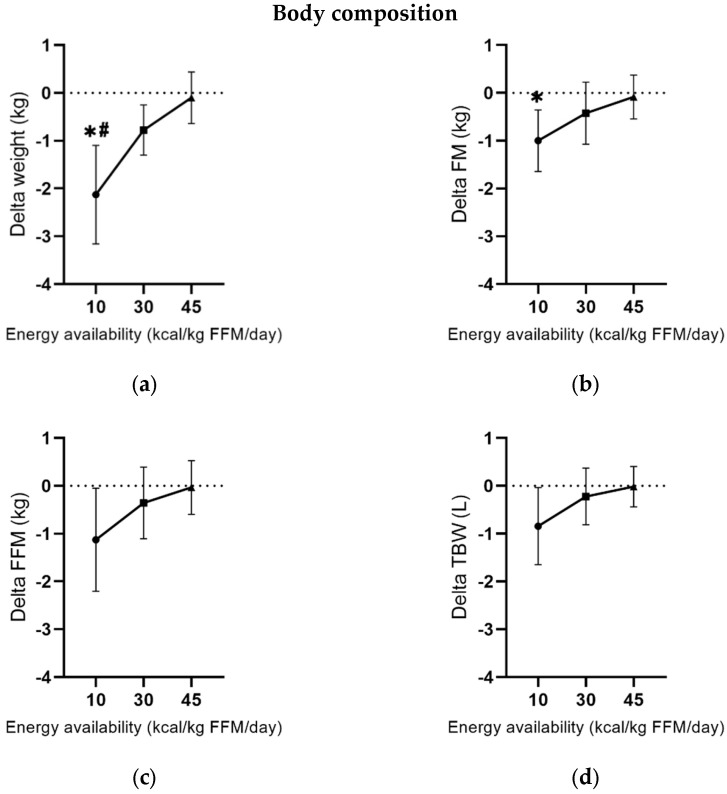
Changes in body composition after five days of different energy availabilities: (**a**) weight in kilogram; (**b**) fat mass in kilogram; (**c**) fat-free mass in kilogram; (**d**) total body water in liter. Values are presented in means ± SD. * *p* ≤ 0.05 with respect to EA45. ^#^ *p* ≤ 0.05 with respect to EA30.

**Figure 3 nutrients-17-00278-f003:**
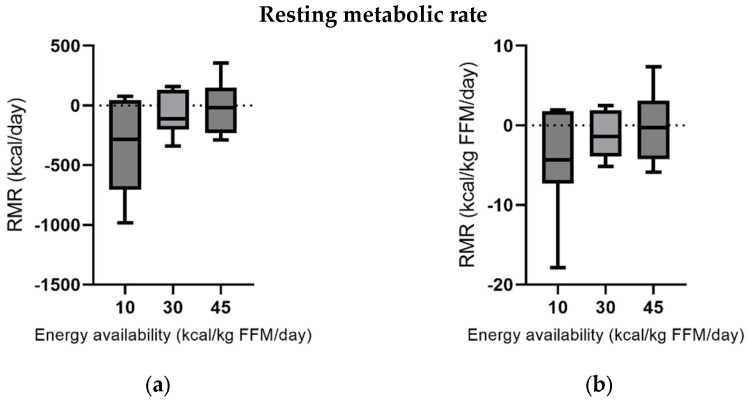
Changes in RMR after energy availability treatment: (**a**) RMR in kcal/day; (**b**) RMR in kcal/kg FFM/day. Values are presented as median (line), first and third quartile (box), and maximum and minimum (whiskers).

**Figure 4 nutrients-17-00278-f004:**
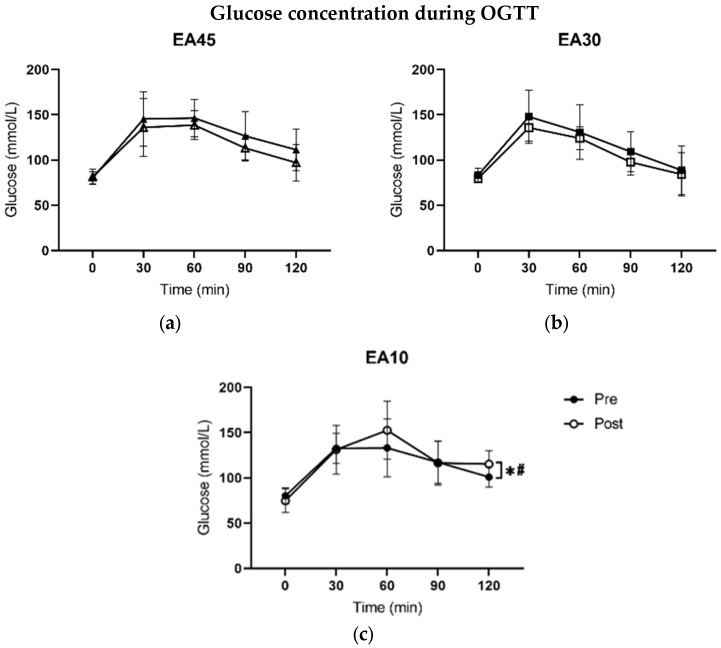
Glucose concentration (mmol/L) during oral glucose tolerance test (OGTT). Capillary blood samples were taken at rest (0) before drinking 75 g glucose and after 30, 60, 90, and 120 min. Lines are separated in pre and post for each group (EA45, EA30, EA10): (**a**) glucose concentration in group EA45; (**b**) glucose concentration in group EA30; (**c**) glucose concentration in group EA10. Values are presented in means ± SD. Significant results contributed by ANOVA from the analysis of delta values are included. * *p* ≤ 0.05 with respect to EA45. ^#^ *p* ≤ 0.05 with respect to EA30.

**Figure 5 nutrients-17-00278-f005:**
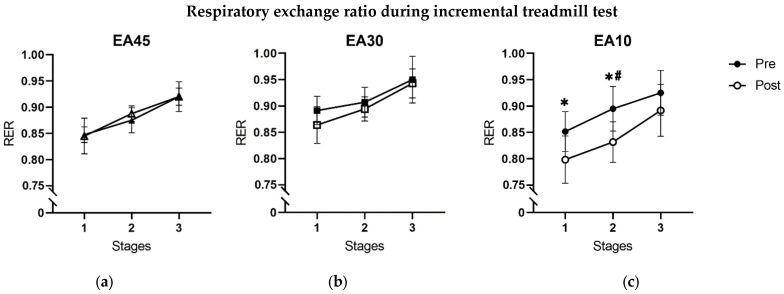
Respiratory exchange ratio (RER) at stages one, two, and three of the incremental treadmill test. The duration of each stage was 5 min. RER was averaged over the last 30 s of each stage. Lines are separated in pre- and post-treatment for each group (EA45, EA30, EA10): (**a**) submaximal respiratory exchange ratio in group EA45; (**b**) submaximal respiratory exchange ratio in group EA30; (**c**) submaximal respiratory exchange ratio in group EA10. Values are presented in means ± SD. Significant results contributed by ANOVA from the analysis of delta values are included. * *p* ≤ 0.05 with respect to EA45. ^#^ *p* ≤ 0.05 with respect to EA30.

**Table 1 nutrients-17-00278-t001:** Experimental treatment effects. Means of energy intake, energy expenditure, and energy availability at baseline and at the end of the intervention (energy availabilities of 45, 30, and 10 kcal/kg FFM/day).

		Baseline (n = 22)	Intervention		
			EA45 (n = 6)	EA30 (n = 9)	EA10 (n = 7)
EI	Kcal/day	2503 ± 693	2704 ± 675	2326 ± 358	964 ± 173 *^#^
	Kcal/kg FFM/day	40.8 ± 4.8	49.6 ± 2.4	35.0 ± 1.8 *	16.4 ± 1.4 *^#^
TEE	Kcal/day	3211 ± 527	2789 ± 503	3356 ± 405	3116 ± 504
	Kcal/kg FFM/day	53.5 ± 6.3	51.9 ± 5.1	50.9 ± 5.0	53.3 ± 5.2
EB: EI–TEE	Kcal/day	−708 ± 459 ^a^	−85 ± 329	−1031 ± 184 *^a^	−2152 ± 376 *^#a^
	Kcal/kg FFM/day	−12.7 ± 8.9 ^a^	−2.2 ± 5.6	−15.9 ± 4.2 *^a^	−36.8 ± 5.0 *^#a^
EEE	Kcal/day	437 ± 328	335 ± 137	436 ± 129	403 ± 81
	Kcal/kg FFM/day	7.3 ± 5.3	6.0 ± 1.2	6.5 ± 1.1	6.9 ± 1.0
EA: EI-EEE	Kcal/day	2067 ± 712	2369 ± 545	1890 ± 262 *	561 ± 101 *^#^
	Kcal/kg FFM/day	33.5 ± 7.1	**43.6 ± 1.7**	**28.5 ± 2.1** *	**9.6 ± 0.5** ***^#^**

Values are presented in means ± SD. EI = energy intake; TEE = recorded 24 h total energy expenditure; EB = energy balance; EEE = exercise energy expenditure; EA = energy availability; bold print: EA achieved according to the groups; * *p* ≤ 0.05 with respect to EA45; ^#^ *p* ≤ 0.05 with respect to EA30; ^a^ *p* ≤ 0.05 comparison between EI and TEE.

**Table 2 nutrients-17-00278-t002:** Delta values of body composition after five days of energy availabilities of 45, 30, and 10 kcal/kg FFM/day.

	Unit	EA45	EA30	EA10	*p*-Value
Body weight	kg	−0.1 ± 0.5	−0.8 ± 0.5	−2.1 ± 1.0 *^#^	<0.001
Fat mass	kg	−0.8 ± 0.5	−0.4 ± 0.7	−1.0 ± 0.7 *	0.038
Fat free mass	kg	0.0 ± 0.6	−0.4 ± 0.8	−1.1 ± 1.1	0.069
Total body water	L	0.0 ± 0.4	−0.2 ± 0.6	−0.8 ± 0.8	0.066

Values are presented in means ± SD. The *p*-value represents the results of ANOVA (interaction effect). * *p* ≤ 0.05 with respect to EA45. ^#^ *p* ≤ 0.05 with respect to EA30.

**Table 3 nutrients-17-00278-t003:** Blood serum levels before treatment, after treatment, and changes in concentrations after five days of controlled energy availability. Values are presented in means ± SD. Pre = before treatment; Post = after treatment; Delta = change score (post-pre). The *p*-value represents the results of the ANOVA (interaction effect).

	Pre	Post	Delta	*p* Value
Glucose (mmol/L)				0.439
45	4.59 ± 0.30	4.57 ±0.21	−0.02 ± 0.14	
30	4.73 ± 0.41	4.52 ± 0.38	−0.21 ± 0.33	
10	4.51 ± 0.26	4.34 ± 0.33	−0.17 ± 0.31	
Triglycerides (mmol/L)				0.026
45	0.67 ± 0.22	0.87 ± 0.34	0.20 ± 0.21	
30	0.86 ± 0.59	0.68 ± 0.23	−0.08 ± 0.19	
10	0.86 ± 0.26	0.69 ± 0.14	−0.17 ± 0.28 *****	
Cholesterol (mmol/L)				0.087
45	3.92 ± 0.74	4.14 ± 0.63	0.22 ± 0.17	
30	4.20 ± 1.30	4.12 ± 1.32	−0.09 ± 0.35	
10	4.35 ± 0.78	4.56 ± 0.85	0.22 ± 0.31	
HDL (mmol/L)				0.272
45	1.70 ± 0.29	1.65 ± 0.38	−0.05 ± 0.14	
30	1.37 ± 0.20	1.32 ± 0.23	−0.05 ± 0.13	
10	1.58 ± 0.40	1.64 ± 0.46	0.06 ± 0.17	
LDL (mmol/L)				0.114
45	2.08 ± 0.79	2.31 ± 0.77	0.23 ± 0.08	
30	2.66 ± 1.22	2.66 ± 1.30	0.00 ± 0.29	
10	2.60 ± 0.54	2.78 ± 0.56	1.19 ± 0.19	
Uric acid (umol/L)				0.007
45	270 ± 70	230 ± 36	−40.8 ± 40.0	
30	273 ± 53	286 ± 61	12.3 ± 15.6 *****	
10	274 ± 65	295 ± 74	21.3 ± 42.6 *****	
Urea (mmol/L)				0.570
45	4.01 ± 0.35	4.26 ± 0.89	0.25 ± 0.98	
30	4.74 ± 1.66	4.63 ± 0.77	−0.11 ± 1.13	
10	4.45 ± 1.19	4.11 ± 1.01	−0.34 ± 0.78	
Creatinine (umol/L)				0.025
45	79.5 ± 18.2	76.3 ± 14.6	−0.03 ± 0.27	
30	82.8 ± 10.8	84.4 ± 11.2	−0.07 ± 0.22	
10	75.5 ± 15.3	82.8 ± 18.0	−0.36 ± 0.20 *****	

***** *p* ≤ 0.05 with respect to EA45.

## Data Availability

The data presented in this study are available upon request from the corresponding author. The data are not publicly available due to confidentiality issues.

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
