# Peer review of "Effects of Short-Term Low Energy Availability on Metabolism and Performance-Related Parameters in Physically Active Adults"

_nutrients, 2025, doi:10.3390/nu17020278_

Round 1

Reviewer 1 Report

Comments and Suggestions for Authors

Dear researchers,

I send my comments, appreciations, suggestions or remarks to your research:

Regarding the development of your introduction, the concerns that you have had for the development of the research are indicated. However, I suggest orienting the narrative to where you really want it to be an area of interest. I point this out to you in this way, because you indicate that this condition is present in athletes, however, before finishing the introduction you indicate that the context you wish to contribute is in public health. Is there any research that addresses areas not related to sports?

Another point of importance is related to the support of this research through Anne's previous studies or works, however, they point out “many reviews” and only present two articles.

Line 81 has the letter [W] in brackets, is this correct?

Materials and Methods

Participants

How were participants recruited, what were the selection criteria?

What information did you collect related to medical history?

Were any instruments or tests applied to confirm the information provided by each participant?

Design

If any software was used to randomize the sample, I suggest reorganizing this section, providing greater clarity in the delivery of the information.

The summary presented in Figure 1 is very interesting.

Experimental protocol, diet, energy expenditure, calculation of energy availability,

I suggest for a better use of the information you provide, subdivide or point out with a subtitle of the variables that will be part of the study. For example, you used BIA to determine weight and body composition, for this, did you use any protocol for the evaluation?

Results

From the positioning to evaluate body composition, men and women are presented separately (line 225). Table 2 suggests the same treatment (it is not in the objective, but, within body composition there are variables that predominate in men and others in women).

Supplementary 3, do men and women have no significant differences between them?

Discussion

Due to the organization of the results, I suggest organizing the discussion according to the titles of the results.

This work contributes to your area, no doubt, but, it does not present strengths, practical applications or future lines of research.

Conclusions

In the introduction you state that you wish to contribute to public health, is this possible?

Author Response

Comment 1: Regarding the development of your introduction, the concerns that you have had for the development of the research are indicated. However, I suggest orienting the narrative to where you really want it to be an area of interest. I point this out to you in this way, because you indicate that this condition is present in athletes, however, before finishing the introduction you indicate that the context you wish to contribute is in public health. Is there any research that addresses areas not related to sports?

Response 1: Thank you for pointing this out. I agree with this comment. Therefore, I have removed the sentence from the introduction as it did not make a recognizable contribution. The statement has led to confusion and now the introduction ends with the hypothesis of the study, which is common. (page: 3, line: 116)

Comment 2: Another point of importance is related to the support of this research through Anne's previous studies or works, however, they point out “many reviews” and only present two articles.

Response 2: Agree. The wording was not well chosen. I have changed it to “previous” to give just a few examples that representatively summarized the topic. I also added two recent reviews for supporting the message. (page: 3, line: 100)

Comment 3: Line 81 has the letter [W] in brackets, is this correct?

Response 3: If a quotation is used within a sentence and the original sentence does not begin with a capital letter, but you place the quotation at the beginning of your sentence, you must put the first letter of the quotation in square brackets to mark the change. Example:

-       Original sentence: “the weather was beautiful yesterday.”

-       Cited sentence: “[T]he weather was beautiful yesterday,”

You have changed the initial letter to make the sentence grammatically correct. This is a generally accepted practice in academic and formal writing.

Comment 4: How were participants recruited, what were the selection criteria?

Response 4: Participants were recruited by addressing students directly in university lectures. Participation was voluntary. The selection criteria were as follows: Sports student, between 18-30 years old, no history of eating disorder or current eating disorder, injury preventing participation in the test battery, healthy (free of illnesses and infections), non-smoker.

The information can be taken accordingly from the methodology section and the results section describing the sample. If additional information is required for a better description, i will be happy to add it.

Comment 5: What information did you collect related to medical history?

Response 5: We had each participant fill out a medical history form before participating. The answers were checked by a physician. Among other things, the following diseases were queried: diabetes mellitus, thyroid disease(s) (e.g. hyperthyroidism, hypothyroidism, enlargement, thyroid surgery, ...), other metabolic diseases, mental illness/s (e.g. anxiety disorder, psychosis, depression, addiction, ...), eating disorder/s (anorexia, bulimia, ...), food intolerance/s, bone bridge, other illnesses, medication intake. If these information are required in the manuscript, i will be happy to add it, but at the moment I can't recognize any request from the comment.

Comment 6: Were any instruments or tests applied to confirm the information provided by each participant?

Response 6: A blood count was taken. Otherwise, the data was not checked via specific tests. With the declaration of consent, the participants agreed to provide all information truthfully. We have no reason to believe that the information provided was a lie, as there was no payment to the participant and participation was voluntary.

Comment 7: If any software was used to randomize the sample, I suggest reorganizing this section, providing greater clarity in the delivery of the information.

Response 7: I agree with this comment. The sample was randomized using Microsoft Excel, with the gender distribution being the only controlled variable. I added the information in the manuscript. (page: 3, line: 136, 137)

Comment 8: The summary presented in Figure 1 is very interesting. Experimental protocol, diet, energy expenditure, calculation of energy availability. I suggest for a better use of the information you provide, subdivide or point out with a subtitle of the variables that will be part of the study. For example, you used BIA to determine weight and body composition, for this, did you use any protocol for the evaluation?
Response 8: I appreciate your positive remarks regarding Figure 1 and am pleased that it is perceived as interesting. Regarding your third comment, I would like to clarify the distinction between the methods used and the parameters collected. As described on page 4, lines 147-149, bioelectrical impedance analysis (BIA) was employed to measure the following parameters: body weight and composition, including fat-free mass (FFM), fat mass, and total body water. I believe that the current presentation, with the methods clearly named in Figure 1 and the subsequent paragraph explaining the specific variables measured, ensures clarity and avoids redundancy. Including the variables directly in the figure might risk overcrowding the visualization and reducing its clarity.
The body composition measurement with the BIA was carried out according to a standardized protocol “Participants had to arrive at 06:30 am in a fasted state (12 hours no eating and drinking) on the first day of pre- and post-testing. We used the results of the InBody 770 system (InBody Europe B.V., Eschborn, Germany) to analyze the measured data (no specific protocol was used).
Unfortunately, I am unsure if I have fully understood your comment correctly, so please feel free to provide any clarifications or additions.

Comment 9: From the positioning to evaluate body composition, men and women are presented separately (line 225). Table 2 suggests the same treatment (it is not in the objective, but, within body composition there are variables that predominate in men and others in women).
Response 9: I appreciate your attention to detail. However, I may not have fully understood your point, as we did not analyze body composition separately for men and women in our study. Due to the small sample size, it was not feasible to statistically test for differences between male and female participants. Only the distribution of women and men (n=11) is shown in line 230. The age, height and weight are shown for the entire group.
Of course, it is well known that body composition generally differs between men and women, particularly in terms of muscle mass and fat distribution. If you were referring to a different aspect, please feel free to clarify further, and I would be happy to address it accordingly.

Comment 10: Supplementary 3, do men and women have no significant differences between them?
Response 10: In Supplementary 3, we intentionally refrained from testing for differences between men and women, as the data presented is purely descriptive. The calculation of energy availability (EA) was conducted without gender-specific adjustments, as it is commonly applied in this manner.
The main purpose of this table was to show that the required energy availability (kcal/kg FFM/day) is actually 45 kcal/kg FFM/day on average (at least in our sample). This confirms current findings of an “normal” healthy energy availability.

Comment 11: Due to the organization of the results, I suggest organizing the discussion according to the titles of the results.
Response 11: I agree with this comment. As the discussion was already structured in the order in which the results were presented, the headings could be inserted for better clarity. (page: 10-13, line: 332, 393, 449, 467)

Comment 12: This work contributes to your area, no doubt, but, it does not present strengths, practical applications or future lines of research.

Response 12: I appreciate your point that my previous discussion did not sufficiently address strengths, practical applications, or future lines of research. I have now added a section with the title "4.4. Strength, Limitations and future directions” and provided more detailed points to better highlight the strengths of the study and potential future research directions. These additions should better emphasize the contributions and future implications of the work. (page: 13-14, line: 467-517)

Comment 13: In the introduction you state that you wish to contribute to public health, is this possible?
Response 13: Thank you for your comment. I appreciate you pointing out this issue, which has been considered across different sections of the paper. Upon reflection, I have clarified that no statements can be made about public health based on the results of this study. As a result, I have removed this point from the overall context and have now answered "no" to this question. It is not possible.

Reviewer 2 Report

Comments and Suggestions for Authors

The study investigates the effects of short-term low energy availability (LEA) on metabolism and performance-related parameters in physically active adults. It provides valuable insights into metabolic adaptations resulting from energy deficits and makes a significant contribution to understanding LEA effects in both men and women. However, there are areas that could be enhanced for clarity, methodological rigor, and overall impact.

Strengths

Relevant Topic: The focus on LEA in both men and women addresses a critical and underexplored area in sports medicine.

Well-Structured Design: The study utilizes a randomized intervention with clearly defined energy availability groups (EA45, EA30, and EA10) and employs a comprehensive battery of tests (e.g., RMR, OGTT, incremental treadmill tests).

Clear Data Presentation: Data are well-documented in tables and figures, offering transparency in results interpretation.

Ethical Conduct: Ethical approval and informed consent processes were adequately followed.

Areas for Improvement

1. Introduction

Issue: The introduction provides a broad overview of LEA but lacks specific research gaps and a sharper focus on the study's novelty.

Recommendation: Clearly highlight how this study expands upon or differs from previous LEA research (e.g., addressing short-term effects or including both sexes).

2. Methodology

Issue:

The use of bioelectrical impedance analysis (BIA) for body composition is mentioned but acknowledged as suboptimal compared to more precise methods like DEXA.

The variability in RMR measurement could have influenced the results.

Recommendation: Acknowledge these limitations explicitly in the methods section. Future studies should consider using gold-standard tools (e.g., DEXA for body composition, doubly labeled water for energy expenditure).

Sample Size: The relatively small sample size (n=22) may limit the generalizability of findings.

Recommendation: Discuss sample size limitations in the discussion or conclusion.

3. Results

Issue: The results are comprehensive but require more emphasis on clinically significant findings.

Recommendation:

Focus on the implications of key outcomes like altered glucose metabolism and reduced RER.

Provide a more detailed explanation of the observed dose-response effects.

4. Discussion

Issue: The discussion adequately interprets findings but could better address contradictory evidence, such as the lack of significant changes in RMR.

Recommendation:

Critically explore why RMR changes were not observed despite evidence of metabolic adaptations.

Relate findings more explicitly to practical implications for athletes and their trainers.

5. Limitations

Issue: While limitations are acknowledged, the impact of menstrual cycle variability and sex differences on outcomes is not sufficiently addressed.

Recommendation: Discuss how hormonal fluctuations in women might have influenced results and propose methods to control for these in future research.

6. Figures and Tables

Issue: Some figures (e.g., Figure 4 on OGTT results) could benefit from more detailed labeling for easier interpretation.

Recommendation: Include clear legends and callouts to highlight significant findings in the graphs.

7. Conclusions

Issue: The conclusions are concise but do not sufficiently elaborate on practical applications.

Recommendation: Include specific recommendations for sports nutrition strategies or training modifications to mitigate the risks of LEA.

The paper is a valuable contribution to the field of sports medicine but requires revisions to address methodological limitations, strengthen the discussion, and enhance the clarity of results presentation.

Author Response

Comment 1:
Strengths:

·       Relevant Topic: The focus on LEA in both men and women addresses a critical and underexplored area in sports medicine.

·       Well-Structured Design: The study utilizes a randomized intervention with clearly defined energy availability groups (EA45, EA30, and EA10) and employs a comprehensive battery of tests (e.g., RMR, OGTT, incremental treadmill tests).

·       Clear Data Presentation: Data are well-documented in tables and figures, offering transparency in results interpretation.

·       Ethical Conduct: Ethical approval and informed consent processes were adequately followed.

Response 1: Thank you for seeing these points as the strength of this study and for elaborating on them once again.

Comment 2:
Introduction

·       Issue: The introduction provides a broad overview of LEA but lacks specific research gaps and a sharper focus on the study's novelty.

·       Recommendation: Clearly highlight how this study expands upon or differs from previous LEA research (e.g., addressing short-term effects or including both sexes).

Response 2: I agree. I could improve this section. I have added a supplementary paragraph which should make the research gaps and the contribution of this study clear: “To address the research gap, this study aims to investigate the short-term effects of LEA in both sexes, which has not been extensively studied before. By including both physically active young women and men, this research seeks to provide a more inclusive understanding of LEA’s impact across sexes. Additionally, this study will explore the immediate metabolic and performance-related adaptations to LEA, thereby expanding upon previous research that primarily focused on long-term effects and sedentary populations.”. The studies by Anne Loucks, which were conducted almost exclusively with sedentary women, and the studies by e.g. Melina et al. (2015), Torstveit et al. (2018), de Souza et al. (2007), Oxfeldt et al., (2023), which examined only one sex, different exposures of LEA or different parameters, are thus considered and summarized. (page: 3, line: 104-110)

Comment 3:

Methodology

·       Issue: The use of bioelectrical impedance analysis (BIA) for body composition is mentioned but acknowledged as suboptimal compared to more precise methods like DEXA.
The variability in RMR measurement could have influenced the results.

·       Recommendation: Acknowledge these limitations explicitly in the methods section. Future studies should consider using gold-standard tools (e.g., DEXA for body composition, doubly labeled water for energy expenditure).
Sample Size: The relatively small sample size (n=22) may limit the generalizability of findings.
Recommendation: Discuss sample size limitations in the discussion or conclusion.

Response 3: We are already aware of the limitations and have tried to present them transparently. They are discussed in the limitations section in the discussion. I have added additional information in the methodology section (page: 4, line: 156) and in the discussion (page: 14, line: 479-495) under 4.4 Strength, Limitations and future directions of the study and have followed your recommendations. I hope that the limitations of the selected methods are appropriately pointed out.

Comment 4:
Results

·       Issue: The results are comprehensive but require more emphasis on clinically significant findings.

·       Recommendation: Focus on the implications of key outcomes like altered glucose metabolism and reduced RER.

Provide a more detailed explanation of the observed dose-response effects.

Response 4: I only partially agree with this comment. I also think it is important to describe the results accurately and clearly, but we want to avoid over-interpreting the results and categorizing them in the results section.

In my opinion, the dose-response relationship is sufficiently described for the body composition values: “Figure 2 indicates a slight dose-response effect: the lower the energy availability, the higher the reduction in body weight, fat mass, FFM, and total body water.” No clear dose-response relationship can be established for the RER values. The graph is presented in incremental steps and not divided into groups as in the case of body composition. Further dose-response relationships could be suspected graphically (e.g. RMR) but are not statistically proven, which is why I do not want to make any statements here. In my opinion, the statement “altered glucose metabolism” should also not be used in the results section, as we only present the results of the OGTT and the submaximal RER values during a step test. The interpretation that this is an altered glucose metabolism belongs in the discussion and is discussed and concluded there accordingly.
In summary, this means that I basically agree with your comment that these results should be emphasized, and we hope that we have succeeded in doing so in the discussion. In the results section, due to the small number of sample size, we would like to avoid over-interpreting the data by highlighting or interpreting results in order to present our measured data transparently.

Comment 5:

Discussion

·       Issue: The discussion adequately interprets findings but could better address contradictory evidence, such as the lack of significant changes in RMR.

·       Recommendation: Critically explore why RMR changes were not observed despite evidence of metabolic adaptations.

Relate findings more explicitly to practical implications for athletes and their trainers.

Response 5: I agree with this comment. I have added a few additions to the RMR measurement/ results both in the main discussion and in the limitations. (page: 11, line: 359-364)
Based on this comment and another reviewer, I have added a paragraph on practical applications at the end of the discussion. (page: 14, line: 510-517)

Comment 6:

Limitations

·       Issue: While limitations are acknowledged, the impact of menstrual cycle variability and sex differences on outcomes is not sufficiently addressed.

·       Recommendation: Discuss how hormonal fluctuations in women might have influenced results and propose methods to control for these in future research.

If any software was used to randomize the sample, I suggest reorganizing this section, providing greater clarity in the delivery of the information.

Response 6: This is a very good point for the discussion, and I would very much like to discuss it thoroughly. Due to the complexity of this topic, which this article does not do justice to, I have opted for a small addition for consideration in future studies. (page: 14, line: 503-506)

If I were to include this topic in full in the discussion, I would have to discuss all of the following points (without addressing basic sex-differences in the hormonal system) and provide sources, which would unfortunately go beyond the scope of this paper.

Hormonal fluctuations in women, particularly those related to the menstrual cycle, can significantly influence physiological and psychological parameters, potentially affecting the outcomes of low energy availability (LEA) studies. These fluctuations can impact metabolism, appetite, energy expenditure, mood, and physical performance, which are all critical factors in LEA research.

Influence of Hormonal Fluctuations

Menstrual Cycle Phases:

·       Follicular Phase: Characterized by lower levels of estrogen and progesterone, this phase may be associated with increased insulin sensitivity and a higher capacity for carbohydrate utilization.

·       Luteal Phase: Marked by elevated levels of estrogen and progesterone, this phase can lead to increased basal metabolic rate, higher body temperature, and changes in appetite and fluid retention.

·       Hormonal Contraceptives: The use of hormonal contraceptives can alter the natural hormonal cycle, potentially stabilizing some fluctuations but also introducing synthetic hormones that can affect metabolism and energy balance.

·       Perimenopause and Menopause: These stages involve significant hormonal changes, including decreased estrogen levels, which can affect body composition, energy expenditure, and overall metabolic health.

Methods to Control for Hormonal Fluctuations in Future Research - Menstrual Cycle Tracking:

·       Regular Monitoring: Gold-standard testing: calender-based, urine-stick (LH), hormone testing in blood serum to control for progesterone.

o   Blood Samples: Collect blood samples to measure levels of key hormones (e.g., estrogen, progesterone, cortisol) and correlate these with study outcomes.

o   Salivary Hormones: Use non-invasive salivary hormone assays for frequent monitoring of hormonal fluctuations.

·       Phase-Specific Testing: Schedule assessments during specific phases of the menstrual cycle (e.g., follicular or luteal) to minimize variability due to hormonal fluctuations.

Standardized Testing Conditions:

·       Consistent Timing: Conduct tests at the same time of day to control for diurnal variations in hormone levels.

·       Controlled Environment: Maintain consistent environmental conditions (e.g., temperature, humidity) to reduce external influences on hormonal responses.

Participant Selection:

·       Homogeneous Groups: Select participants with similar menstrual cycle characteristics (e.g., regular cycles, non-users of hormonal contraceptives) to reduce variability.

·       Stratified Analysis: Analyze data within these subgroups.

Longitudinal Studies:

·       Extended Monitoring: Conduct longitudinal studies to observe changes over multiple menstrual cycles, providing a more comprehensive understanding of hormonal influences on LEA.

·       By implementing these methods, future research on low energy availability can better account for hormonal fluctuations in women, leading to more accurate and reliable results.

I added information for the explanation for the randomization. (page: 3, line: 136, 137)

Comment 7:
Figures and Tables:

·       Issue: Some figures (e.g., Figure 4 on OGTT results) could benefit from more detailed labeling for easier interpretation.

·       Recommendation: Include clear legends and callouts to highlight significant findings in the graphs.

Response 7: I acknowledge the comment and regret that the graphical presentation may not have been sufficiently clear. However, all relevant statistical markers and legends indicating group assignment and significant differences are indeed provided. Asterisks (*) and hashtags (#) are used to denote significant differences compared to the respective conditions, as described in the figure legend. I have checked all the figures again for completeness. The other reviewers did not raise these concerns, so I will refrain from making any adjustments for now. I appreciate your input and will keep it in mind moving forward.

Comment 8:
Conclusions

·       Issue: The conclusions are concise but do not sufficiently elaborate on practical applications.

·       Recommendation: Include specific recommendations for sports nutrition strategies or training modifications to mitigate the risks of LEA.

Response 7: As described in a comment above, I have added a paragraph at the end of the discussion on practical applications. I hope the additions are consistent with the claims in this comment and provide recommendations for sports nutrition strategies or training modifications to mitigate the risks of LEA. Unfortunately, I cannot make more detailed recommendations based on my findings as this was an RCT and not a systematic review. So primarily the results from this study are concluded.

In summary, I would like to thank you for compiling these comments and feedback. They were all very helpful in improving the article, even if not all of them were included in the final version. I hope the explanations complement the current state of affairs, and for any remaining open points, I would appreciate further feedback if they could not be adequately addressed.

Reviewer 3 Report

Comments and Suggestions for Authors

From my point of view, the study conducted by Nolte et al. can be considered for publication in the Nutrients journal after some alterations. Here are my recommendations:

In the abstract, please add some directions for further investigations.

“nutritional considerations for athletes” is not an adequate keyword. Here you have 4 words (almost a sentence), please revise it.

Lines 99-100: “Many reviews recommend further research on the metabolic effects of LEA in different populations and conduct both short-term and long-term studies” – Please, provide more references to support this. Reference 20 has seven years, please provide more recent references here.

Lines 116-117: “Thirty young, active, and healthy women and men were recruited for the present study, while twenty-two successfully completed it.” – Why this sample size? Please, justify it and explain how it can be considered representative. This should also be moved to the Results section.

The sample size should be discussed as a study limitation. I believe this section should be provided separately from the Discussion.

Conclusions could be improved and directions for further studies should be given.

Author Response

Comment 1:
In the abstract, please add some directions for further investigations.

Response 1: Thank you for the comment. I agree and have added a corresponding sentence in the abstract.
“Future research should focus on longer exposures of LEA and sex-specific comparisons (including the menstrual cycle) on LEA symptoms.” (page: , line: )

Comment 2:
“nutritional considerations for athletes” is not an adequate keyword. Here you have 4 words (almost a sentence), please revise it.

Response 2: I can well understand the comment and changed it to "athlete nutrition." This should represent the field of nutrition in the athlete population. (page: 1, line: 28, 29)

Comment 3:

Lines 99-100: “Many reviews recommend further research on the metabolic effects of LEA in different populations and conduct both short-term and long-term studies” – Please, provide more references to support this. Reference 20 has seven years, please provide more recent references here.

Response 3: I agree. I have already corrected the sentence based on another comment and have now added two more reviews (Skarakis et al., 2021 & Melin et al., 2024) to support the statement and ensure its recent relevance. (page: 3, line: 100, 101)

Comment 4:
Lines 116-117: “Thirty young, active, and healthy women and men were recruited for the present study, while twenty-two successfully completed it.” – Why this sample size? Please, justify it and explain how it can be considered representative. This should also be moved to the Results section.

Response 4: I only partially agree with this comment. I have added a short addition to the 2.1. Participants paragraph but would leave the description in the methods section. (page: 3, line: 119, 120)

I also have the following addition for the reviewer:

We did a power-analysis before.
The optimal sample size was calculated by using a power analysis with comparative data from another study with the respiratory quotient (RQ) as main parameter. Subjects performed a ramp treadmill test to determine V’O2max and RQ at 60% V’O2max after 5-days fasting period (Eibl et al., 2015). With an effect size of d=0,986 (Eibl et al., 2015), which is a large effect according to interpretation of d to Cohen's (1988) and converted into η2 = 0.14, and a power of 0.9, one would need a total of 33 subjects to obtain a significant result with a repeated-measures ANOVA with 3 groups and 2 measuring time points (α = 0.05).

Unfortunately, the test phase could only be started with 30 participants after the recruitment phase.

Eibl, A., Limmer, M., Hentz, C., Krusche, T., Felker, K., Sonnefeld, H., & Platen, P. (2015). The effects of a 5-day fasting period on endurance related parameters in healthy adults. Conference: Annual Congress of the EUROPEAN COLLEGE OF SPORT SCIENCE, At: Malmö, Sweden.

Comment 5:

The sample size should be discussed as a study limitation. I believe this section should be provided separately from the Discussion.

Response 5: Based on the other reviewers, I have added a paragraph in the discussion (4.4. Strength, Limitations and future directions). There, in the limitations, I also addressed the concerns regarding the samples size. I hope that I can fulfill the comment with the appropriate additions to the text. (page: 13, 14, line: 492-494)

Comment 6:

Conclusions could be improved and directions for further studies should be given.

Response 6: Based on the other reviewers, I have added a chapter 4.4 Strengths, limitations and further directions at the end of the discussion. Future recommendations for studies in this area should now be comprehensively explained there. To avoid repetition, I would refrain from mentioning them in the conclusion. (page: 14, line: 496-509)

In summary, I would like to thank you for compiling these comments and feedback. They were all very helpful in improving the article, even if not all of them were included in the final version. I hope the explanations complement the current state of affairs, and for any remaining open points, I would appreciate further feedback if they could not be adequately addressed.
